# A Novel Change Detection Approach Based on Spectral Unmixing from Stacked Multitemporal Remote Sensing Images with a Variability of Endmembers

**Ke Wu [1],\*, Tao Chen [1] , Ying Xu [2,3], Dongwei Song [1] and Haishan Li [1]**

[1] Institute of Geophysics and Geomatics, China University of Geosciences, Wuhan 430074, China; taochen@cug.edu.cn (T.C.); 1201610414@cug.edu.cn (D.S.); haishanli@cug.edu.cn (H.L.)

[2] National Satellite Ocean Application Service, Ministry of Natural Resources, Beijing 100081, China; xuying@mail.nsoas.org.cn

[3] Key Laboratory of Space Ocean Remote Sensing and Application, Ministry of Natural Resources, Beijing 100081, China

\* Correspondence: kewu@cug.edu.cn; Tel.: +86-136-3862-2685

**Abstract:** Due to the high temporal repetition rates, median/low spatial resolution remote sensing images are the main data source of change detection (CD). It is worth noting that they contain a large number of mixed pixels, which makes adequately capturing the details in the resulting thematic map challenging. The spectral unmixing (SU) method is a potential solution to this problem, as it decomposes mixed pixels into a set of fractions of the land covers. However, there are accumulated errors in the fractional difference images, which lead to a poor change detection results. Meanwhile, the spectra variation of the endmember and the heterogeneity of the land cover materials cannot be fully considered in the traditional framework. In order to solve this problem, a novel change detection approach with image stacking and dividing based on spectral unmixing while considering the variability of endmembers (CD_SDSUVE) was proposed in this paper. Firstly, the remote sensing images at different times were stacked into a unified framework. After that, several patch images were produced by dividing the stacked images so that the similar endmembers according to each land cover can be completely extracted and compared. Finally, the multiple endmember spectral mixture analysis (MESMA) is performed, and the abundant images were combined to produce the entire change detection thematic map. This proposed algorithm was implemented and compared to four relevant state-of-the-art methods on three experimental data, whereby the results confirmed that it effectively improved the accuracy. In the simulated data, the overall accuracy (OA) and Kappa coefficient values were 99.61% and 0.99. In the two real data, the maximum of OA were acquired with 93.26% and 80.85%, which gained 14.88% and 13.42% over the worst results at most. Meanwhile, the Kappa coefficient value was consistent with the OA.

**Keywords:** spectral unmixing; change detection; stacked images; endmember

## 1. Introduction

As is known, change detection (CD) is the main application of remote sensing technology, which is the process of analyzing changes of surface features with multitemporal remote sensing imageries in the same area [1]. Sensors with median spatial resolutions, such as Landsat TM and Sentinel, can always be applied to different CD techniques to capture changes, because they always have high temperatures for appropriate areas. Therefore, to characterize, model and analyze the changes from the median/low spatial resolution, remote sensing images are very meaningful. Some learning-based approaches, just like the post-classification comparison (PC) [2], the change vector analysis (CVA) [3], support vector machine (SVM) [4], decision trees (DT) [5], Markov random field (MRF) [6], augmented linear mixing model (ALM) [7], etc., have been known and used in the past several

decades. They utilize the empirical rules or learned strategies to create feature difference maps. A simple method is to calculate the feature difference maps and separate changing areas by thresholding. These kinds of techniques always assume that a pixel belongs to a single class, and the determination of change or no-change is considered under a "full-pixel "condition, which means that the traditional used methods may have to analyze this issue at the pixel level [8]. However, different materials can jointly occupy a single pixel due to the limitation of the spatial resolution; thus, lots of mixed pixels occur widely in the images. If a mixed pixel can be regarded as a combination formed by several different classes rather than a single class, then a loss of information and low CD accuracy are inevitable during such a "full-pixel-level" CD process [9,10].

The spectral unmixing (SU) technique is regarded as one of the most important means to solve a problem by giving an abundance of images of surface cover classes constituting the area of a pixel [11,12]. The process can be described as follows. Firstly, different endmember spectra are extracted from the multitemporal images. After that, the abundances of each land-cover are estimated, and the differential proportions are compared to obtain the final change map. In this way, the change detection process is performed from the "full-pixel level" to the "sub-pixel level", which can be called CD_SU. The performance of CD_SU has been proven better than that of the traditional CD methods [13]. Therefore, this processing method has been receiving more attention with its unique advantages. It is necessary to develop the advanced techniques in CD studies for analyzing sub-pixel-level spectral changes. In references [14,15], a linear spectral mixture analysis was proposed for estimating impervious surface distributions by addressing the differencing fraction images. In reference [16], a hyperspectral mixture analysis was developed in the multitemporal images for feature selection for the species mapping. In reference [17], different combinations in the multitemporal images were effectively explored in the sub-pixel-level CD approach and improved the result accuracy. In reference [18], a change vector analysis model was integrated in the CD_SU model to reduce the effect of the cumulative error in a post-classification comparison. Moreover, it was closely related to some image processing technologies, such as sub-pixel mapping [19], target detection [20], feature information extraction [21], data fusion [22], etc.

Although great progress has been made in the research, the CD problem in multitemporal images was addressed from the spectral unmixing point with supervised techniques. There are accumulated errors in fractional difference images that can produce lots of false change and noises. It is hard to choose a suitable threshold value to identify the changes, which makes the adequate capturing of details in fractional differential images challenging and leads to poor change detection results [23]. Some researchers have proposed that the spectral signatures of the multitemporal images were considered to be a stacked feature space, and changed and unchanged classes can be analyzed in a uniformed unsupervised framework dealing with the issue [24,25]. Thus, the accumulated errors are effectively reduced, and this format makes it easy for us to find the typical changed land cover type [26]. However, small spectral changes usually occur and need to be regarded as a potential land cover change in stacked image data. Meanwhile, land cover spectral properties may show a high variability due to the inevitable external conditions, such as sunlight conditions and seasonal changes, etc. [27,28]. If this situation is not considered, the atypical endmembers' spectral signatures cannot represent the real abundance images and, thus, decrease the separability. In order to overcome the mentioned problem, several patch images are produced by dividing stacked images so that similar endmembers, including changed and unchanged land cover types, can be completely extracted and compared. The optimal endmember matrix is generated, and a multiple endmember spectral mixture analysis (MESMA) is performed. Thus, the variability of the base endmembers according to each land cover can be taken into full consideration.

In view of this way of thinking, a novel change detection approach with image stacking and dividing based on spectral unmixing while considering the variability of endmembers (CD_SDSUVE) is proposed in this paper. Firstly, the spectral signatures

of the multitemporal images are considered to be a stacked feature space. Thus, the considered CD problem is transformed to analyze the spectral variations within a single pixel. After that, the stacked image is divided into several small patch images, to which the endmember groups are built, and combined into an integrated endmember pool. The optimum endmember combinations associated with the changed and unchanged endmember types are identified in the pool. In this way, the endmember variability is addressed by allowing the endmember number and type to vary at the pixel level. Finally, all of the changed and unchanged regions are unmixed based on the MESMA. The proportions of the different endmember types are compared, and the maximized one is taken to obtain the final CD results.

The organization of this article is as follows. Section 2 describes the relevant background of CD_SU. Section 3 presents the proposed CD_SDSUVE, pointing out the main properties of the stacking, dividing and endmember with a variety of domains and their roles in CD. Section 4 describes the experimental images and analyzes and discusses the CD results. Section 5 is the computational complexity analysis. Section 6 draws the conclusions of this work.

## 2. Relevant Background

For the past few years, CD_SU have been used in multitemporal remote sensing images and achieved good results. The general approach is based on two processes. In the first process, the mixed pixels of the images with different periods are decomposed, respectively, to obtain the land cover abundances. Due to the simple computation and clear physical interpretation, the linear mixture model (LMM) has been widely used, which assumes that the observed spectra are the linear combinations of endmembers weighted by their corresponding abundances, and distinct endmembers are independent from each other [29]. The spectral signature of a mixed pixel is expressed as:

$$x = \mathbf{S}\alpha + n \tag{1}$$

where $x$ is the spectral vector value of a mixed pixel, $\mathbf{S}$ is the built endmember matrix, $\alpha$ is the abundance column vector according to each endmember and $n$ is the noise. The least squares (LS) method is utilized to obtain the most suitable abundance $\alpha$. Moreover, there are two important qualifications to the abundances, nonnegative and sum-to-one [30,31]. Under this constraint, the abundances according to every land cover type are significant. If the number of endmembers in the image is M, then the above Equation (1) can be described as:

$$x = \sum_{i=1}^{M} s_i \alpha_i + n \tag{2}$$

where $s_i$ and $\alpha_i$ are the $i$th endmember spectral and corresponding abundance values. In the second process, the different images are produced by the abundance of different land cover types, and the binary image is regarded as the CD map. During the process, a comparison is performed for each pixel pair in the multitemporal images to generate the initial proportional differences of each class. For example, if $\alpha_{k,t1}$ and $\alpha_{k,t2}$ are the abundance values at time t1 and t2 for the $k$th endmember types, then the abundance differences value can be defined as $\left| \alpha_{k,t1} - \alpha_{k,t2} \right|$. In general, the distribution of changed and unchanged classes is different in the difference image histogram. The former ones are always spread out over the sides of histogram, and the latter are mainly gathered in the center [32,33]. Therefore, the threshold value can be set as twice the standard deviation of the unchanged land cover distribution. With an appropriate thresholding value based on the experience, the changed area is achieved from the correlated resultant image.

## 3. The Proposed CD_SDSUVE Algorithm

Different with the traditional CD_SU method, we use image stacking, blocking and consider variability of endmembers in different patches, where the most suitable endmem-

ber combination is used in the multiple endmember spectral mixture model, to obtain the CD map.

### 3.1. Stacking and Patch Generation of the Multitemporal Images

In order to intuitively describe the changed classes and avoid process errors, image stacking is considered in this study. The multitemporal images are stacked in the uniformed framework. The change land cover type can be directly judged by the combined spectral curve in the stacked image, which is more intuitive, and the spectral features are easy to be captured. The conventional process of constructing the differential image is replaced by selecting the changed spectrum of the stacked image, which makes the proposed algorithm more practically useful. For example, if there is a change for the pixels, the first half and the second half of the pixel spectrum seem to be significantly different. On the contrary, they have strong similarities when there is no changed for the pixels. After that, the stacked image is divided into several image patches that highlight the small endmember features. In this situation, the changed and unchanged endmember spectral signatures can be sufficiently analyzed in each small block. The image patch size is much smaller than the entire image after the dividing process. The patch scheme simultaneously handles both the issue of a possible large number of endmembers and small local spectral variability effects. Although the number and type of endmembers are increased, the redundant endmembers generated can be merged or eliminated later. In this case, the relative change of deformation can be fully considered, which effectively reduces the errors of the endmember extraction. If M1 and M2 are the two remote sensing images at different times for CD (see Figure 1), then M1 and M2 can be stacked into a MS pile. If the divided scale is defined as s, then MS is divided into p ($p = s^2$) regularly shaped patches. MS-p is the *p*th patch of MS, which is dominant. Note that the parameter s is defined depending on the size of the image and the significance of the occurred change targets in the scene.

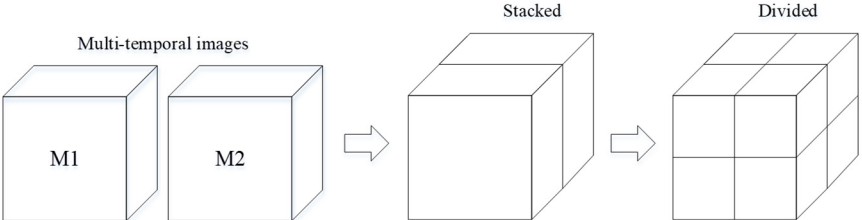

**Figure 1.** The illustration of the Multitemporal images stacking and dividing.

### 3.2. The Initial Endmember Group Construction for Small Patch Image

Firstly, the principal component analysis (PCA) is adopted to estimate the number of endmembers in the small patch image, because it is based on variant information and more effective in multispectral images [34]. After that, the initial endmember spectra signatures E of each patch image are identified by the N-FINDR algorithm, since it is robust and universal [35]. This procedure does not require any input parameters. The set of pixels that define the simplex with the maximum volume was found within the dataset. First, a dimensionality reduction in the original image is accomplished by using the minimum noise fraction (MNF) transform. Next, randomly selected pixels qualify as endmembers, and a trial volume is calculated as follows. The matrix of the endmembers is defined as E, and a row of unit elements is added to construct an augmented matrix:

$$E = \begin{bmatrix} E_1 & E_2 & \cdots & E_m \\ 1 & 1 & \cdots & 1 \end{bmatrix} \tag{3}$$

where $E_i$ is the *i*th endmember spectrum, and m is the defined endmember number of the dimensions occupied by the data. The volume of a constructed simplex using the purest pixels is assumed the largest. Then, the algorithm begins with randomly selected

pixels, and the volume V of the formed simplex according to the specified endmembers is calculated.

$$V = \frac{abs(|E|)}{(m-1)!} \tag{4}$$

The process is repeated for every pixel in the image. If the recalculated volume is increased, the new endmember is replaced. This procedure continues until no more replacements are done in the execution [36].

### 3.3. Construction of the Endmember Pool for the Stacked Image

In this section, the differentiation and combination of the endmember spectra are completed in the blocks, and the endmember pool is constructed. For the sake of description, the divided scale s is defined as 2. There are 4 patches, and the initial endmember groups $G_1 \sim G_4$ are performed in Figure 2.

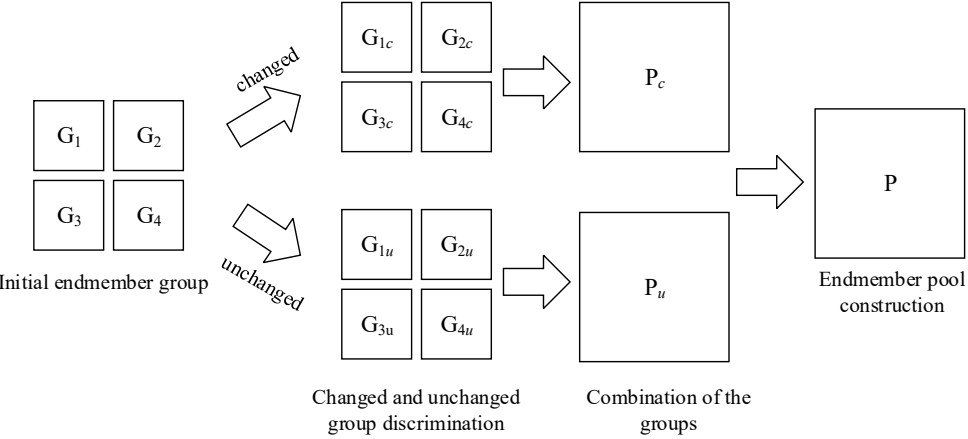

**Figure 2.** The process of the endmember pool construction.

Firstly, the difference value d is defined to discriminate the changed land cover classes in the endmember group. The Euclidean distance of the spectral differences is analyzed according to a change vector analysis as follows:

$$d = \sqrt{\sum_{b=1}^{B} (s_{b+B} - s_b)^2} \tag{5}$$

where $s_b$ and $s_{b+B}$ are the *b*th and *(b + B)*th channel in the stacked image, representing the former and latter part. *B* is the number of the spectral channels. Therefore, the changes are associated with the high values, and no changes are associated with the low values. An appropriate threshold T is automatically set based on the histogram of the B-dimensional difference image computed by subtracting the multitemporal images pixel-by-pixel. Since the intensity change values often follow the Gaussian mixture model, the expectation maximization (EM) algorithm and Bayes discriminant criterion are used to estimate the threshold T [37,38]. If d is larger than T, it is considered significant, indicating a certain change. Otherwise, it is not. According to the given rule, the *i*th patch image endmember group can be defined as $G_i = \{G_{ic}, G_{iu}\}$ and I = [1, 4], where $G_{ic}$ and $G_{iu}$ are the changed and unchanged groups, respectively. Secondly, a comparison should be implemented to combine the endmember spectra in each separated changed or unchanged endmember group. The defined rule is obviously the key factor in the procedure. Due to the explicit physical meaning, the spectral angle mapper method (SAM) technique is used

to discriminate the similarity by calculating the angle [39]. The formula for calculating the spectral angle $a$ between the two specified endmembers is Equation (6):

$$a(e_s, e_t) = \arccos\left(\frac{e_s^T e_t}{\sqrt{e_s^T e_s}\sqrt{e_t^T e_t}}\right) \tag{6}$$

where $e_\text{s}$ and $e_\text{t}$ are the two spectra representing the specified endmembers. The smaller the angle is, the more similar the two spectra are. If the two endmembers are less than the empirical threshold, they belong to the same type. In this way, all of the possible endmembers' spectra are classified and labeled. Thus, all of the changed groups $G_{1c}{\sim}G_{4c}$ and unchanged groups $G_{1u}{\sim}G_{4u}$ are combined into separated endmember pools $P_c$ and $P_u$, respectively. The endmember pool P is constructed based on the above process.

### 3.4. Multiple Endmember Spectral Mixture Analysis

Due to the integration of several different groups, one class may correspond to several different endmember spectral signatures in the endmember pool P. Therefore, the variability of endmembers should be considered. The MESMA, which is the extension of LMM, is adopted to obtain the abundance image of the changed land cover types and generate the CD result. The main idea is to exhaust all endmembers in the endmember pool P and make sure the endmember spectrum for each class is the most suitable. However, a mass of redundant computation arises when there is a lot of candidate endmember spectra. Especially, the blocking easily caused many similar spectra according to the same land cover type. In this case, the endmember average root mean square error (EAR) indicator is proposed to optimize the selection of the endmember spectrum [40,41]. The EAR can determine the average error of an endmember modeling spectra within its land cover type. The minimum EAR endmember is the most representative endmember for a land cover class within several similar spectra. For example, there are N candidate endmembers' spectra $\left\{E_i^1, E_i^2, \ldots, E_i^N\right\}$ for the $i$th class, and the $EAR_i^s$ indicator is as follows:

$$EAR_i^s = \frac{1}{N-1}\sum_{t=1}^{n-1} RMSE(E_i^s, E_i^t) \tag{7}$$

where $RMSE(E_i^s, E_i^t)$ is the average of the root mean square error between the $E_i^s$ and $E_i^t$, and $n$ is the number of endmembers for a class. The lower the $EAR_i^s$, the better the representation of the spectrum. If it is high, the spectrum may be outlines and not representative. The other endmembers' signatures for the rest of the land cover types are confirmed based on the above thought. In this way, all of the possible endmembers' spectra are combined to form a variable endmember matrix for the stacked image. The candidate pixel is decomposed according to the specified endmember matrix. The MESMA model is revised based on Equation (2) and shown below.

$$x = \sum_{i=1}^M \sum_{j=1}^{D_i} qs_{ij}\alpha_{ij} + n \tag{8}$$

where M is the number of the endmember, $D_i$ is the number of spectral according to the $i$th land cover type, $s_{ij}$ and $\alpha_{ij}$ are the $j$th endmember spectra in the $i$th class type and the corresponding abundance value, respectively, q is the label (0 or 1) representing whether the concerned endmember spectral is used, and n is the noise. Like the LMM, the MESMA also has two constraints: $0 \leq s_{ij} \leq 1$ and $\sum_{i=1}^M \sum_{j=1}^{D_i} qs_{ij}\alpha_{ij} = 1$. Finally, the considered CD problem is formalized as to estimate the abundance of changed classes within a single pixel and generate an entire result. The summary of the CD_SDSUVE algorithm is described in Table 1. The whole process of the CD_SDSUVE methodology is described in Figure 3.

**Table 1.** The CD_SDSUVE methodology.

| **Input: Remote Sensing Data: M1 and M2** |
|---|
| Step 1: Two data M1 and M2 are stacked to a new data MS |
| Step 2: Divided scale s is defined, and MS is divided as p ($p = s^2$) patch images |
| Step 3: Initialize the endmember group $G_i$ for ith patch image by N-FINDR |
| Step 4: Analyze each endmember group and generate a whole endmember pool P for the stacked image |
| (1) Discrimination of the changed and unchanged land cover types $G_i = \{G_{ic}, G_{iu}\}, i \in [1, p]$ |
| (2) Combination and analysis of the similar endmember spectrum by SAM |
| (3) The endmember pool P is built based on all of the endmember groups $G_i$ |
| Step 5: Construction of multiple endmember spectral unmixing model |
| (1) Select the suitable endmember class using EAR indicator |
| (2) Spectral unmixing with the multiple endmember matrix |
| Step 6: compare the abundance and the final change map is generated |
| Output: The change map of M1 and M2 |

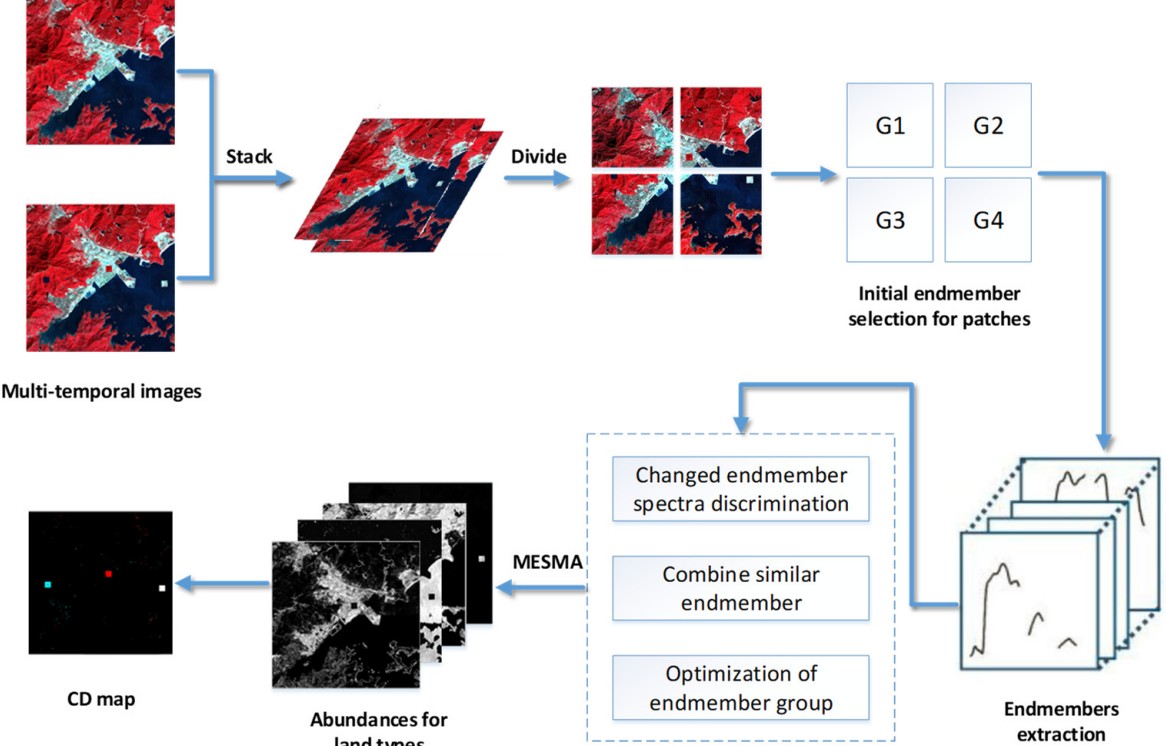

**Figure 3.** The process of the CD_SDSUVE methodology.

## 4. Experiments and Analysis

In order to prove the feasibility of the method and steps, three different remote sensing images were, respectively, presented in this section. Firstly, a simple simulated Landsat TM image to evaluate the performance of the proposed algorithm. To produce the simulated changed situation, small areas were cut from the original image and exchanged as different changes so that the result can be compared with the one from the original image. Secondly, two pairs of multitemporal images were chosen to test the performance of the algorithm in real scenarios. The proposed CD_SDSUVE was tested and compared with four different CD methods, including CD based on the post-classification (CD_PC), CD based on spectral unmixing (CD_SU). CD with image stacking based on spectral unmixing (CD_SSU) and CD with image stacking and dividing based on spectral unmixing (CD_SDSU). These algorithms were implemented several times just to keep them fair. Under the same parameters,

there were very small variations according to their results. Moreover, the experimental results were checked by visual and statistical accuracy assessments. The percent correctly classified (PCC) and the Kappa coefficients of agreement were calculated to evaluate and test the change detection precision of these presented five methods.

### 4.1. Simulated Multitemporal Image Data

The spatial resolution of the Landsat TM was 30 m with six bands. The wavelength mainly ranged from visible to near-infrared. A simple image with $400 \times 400$ referring to the urban area of Shenzhen, Guangdong, China was chosen to carry out the experiment (Figure 4a). Through the visual interpretation, there were mainly three classes: water, vegetation and urban, defined as W, V and U, respectively. To produce the simulated changed situation, three $20 \times 20$ square areas regarded as the water, vegetation and urban classes were cut from the original image (Figure 4b). They were exchanged and represented as three different changes, described as vegetation-to-water (VW), urban-to-vegetation (UV) and water-to-urban (WU). Three combined spectral curves are shown in Figure 4c, whose spectral signatures include two components regarded as changed. Green is UV, red is VW and blue is WU.

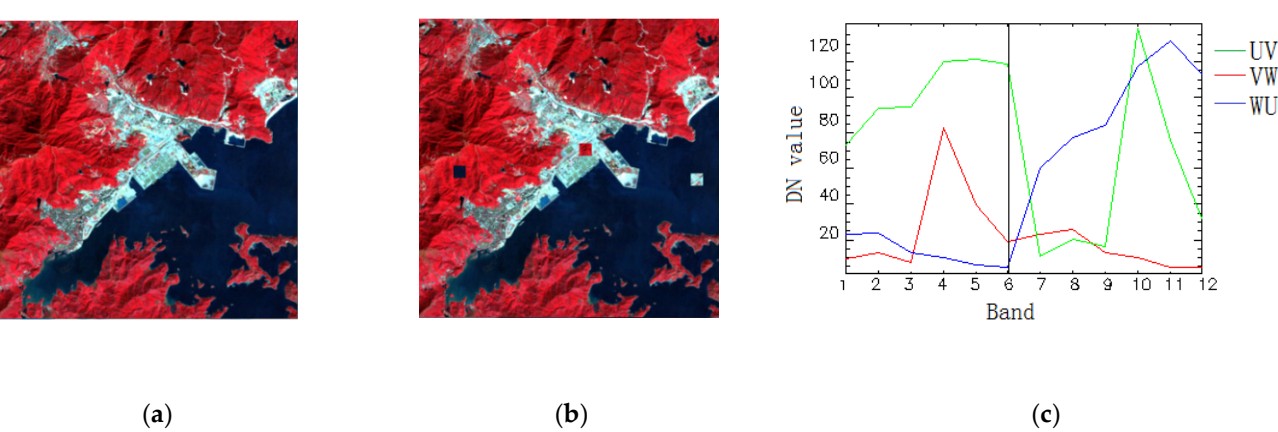

(**a**)                                        (**b**)                                        (**c**)

**Figure 4.** Simulated Multitemporal image data: (**a**) the original TM image (wavelength: R: 860 nm, G: 655 nm and B: 560 nm); (**b**) simulated TM image with three land cover changes and (**c**) three changed endmember spectral signature descriptions.

In view of the image size and the simple distribution of the land cover types, the divided scale s was defined as 2. Four endmember groups: $G_1$, $G_2$, $G_3$ and $G_4$ were generated. The number of endmembers for the groups was set as nine, nine, seven and nine, respectively. Thus, all possible spectra according to the different land cover types in each group were extracted, and the numbers are listed in Table 2. For example, there were two similar spectra according to class W, three similar spectra expressed according to class U, three similar spectra according to class V and only one spectral to class VW in the endmember group $G_1$. Meanwhile, there was no appropriate spectrum according to classes WU and UV, which was defined as zero (NULL). It means these two endmember classes were not included. The endmembers signatures were extracted by the N-FINDR algorithm. The changed and unchanged endmember spectra were discriminated by Equation (5) with a threshold 0.47. The similar endmember spectra were combined by SAM with 0.1. The endmember pool was constructed, and the EAR optimization was performed to select the ultimate representative spectra for different land cover types. Using Equation (8), the MEMSA model using different endmember combinations was applied to the entire stacked image, representing the six actual endmember types included. The acquired abundance images for each class are shown in Figure 5a–d. The higher the abundance value, the brighter it was and vice versa. They provided apparent proportions of the six land covers: W, V, U, VW, UV and WU. The maximum value of each pixel of all the abundance images was set as the determinate category, and the final entire change classification map was constructed. The reference land cover change map is shown in

Figure 6a, and the change detection results by different methods, CD_PC, CD_SU, CD_SSU, CD_SDSU and CD_SDSUVE, are shown in Figure 6b–e, respectively.

**Table 2.** The possible endmember spectral numbers in each group.

| Group \ Class | Number | | | | | |
|---|---|---|---|---|---|---|
| | **W** | **U** | **V** | **WU** | **UV** | **VW** |
| $G_1$ | 2 | 3 | 3 | 0 | 0 | 1 |
| $G_2$ | 1 | 3 | 2 | 0 | 3 | 0 |
| $G_3$ | 2 | 3 | 1 | 0 | 0 | 1 |
| $G_4$ | 1 | 3 | 2 | 3 | 0 | 0 |

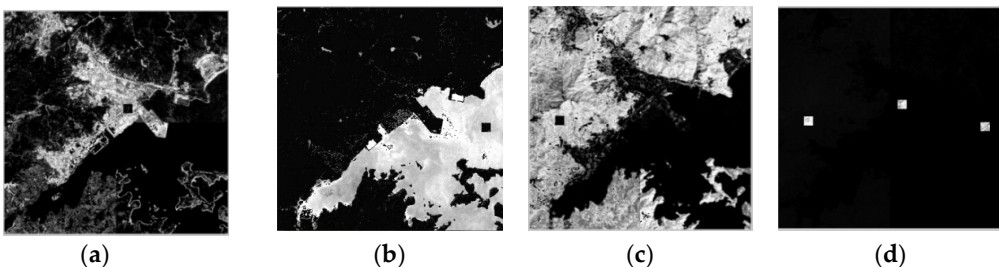

**Figure 5.** The abundances in the entire image: (**a**) urban (U); (**b**) water (W); (**c**) vegetation (V) and (**d**) all of the changed classes (VW, UV and WU).

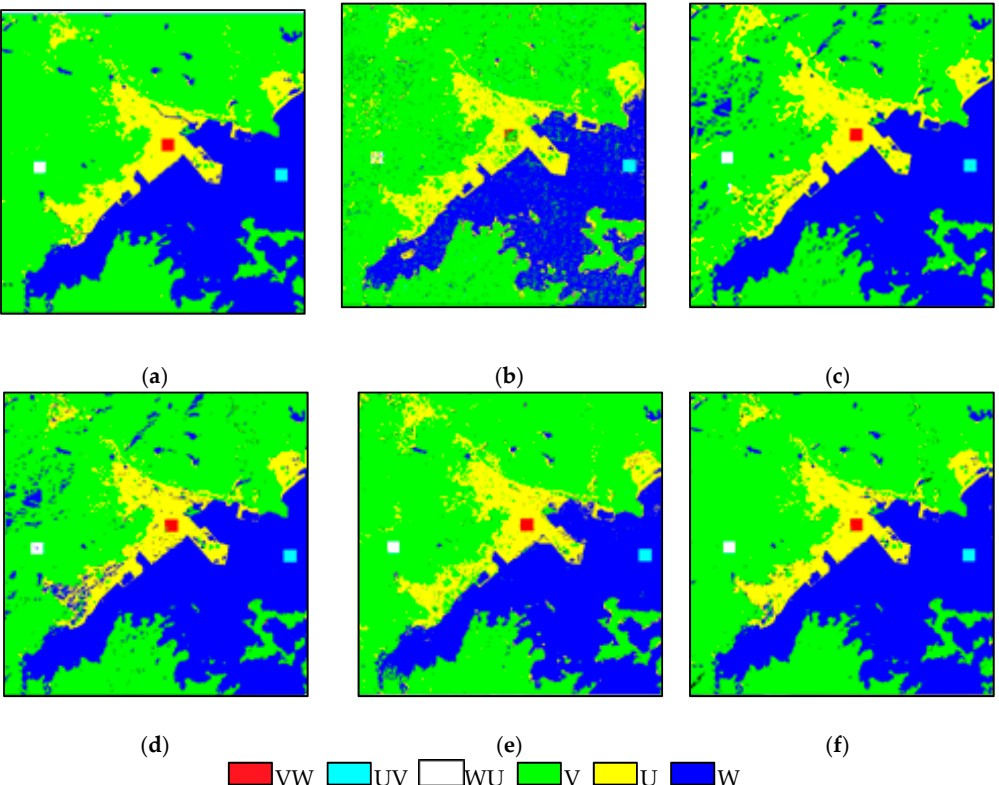

**Figure 6.** The change detection results of the simulated data: (**a**) reference image, (**b**) CD_PC, (**c**) CD_SU, (**d**) CD_SSU, (**e**) CD_SDSU and (**f**) CD_SDSUVE.

The reference land cover change map in Figure 6a is produced by these manmade exchanged areas. The VW is shown in red. The UV is shown in cyan, and the WU is shown in white. Through the visual comparison to the reference map, CD_PC is the worst

result. There are many isolated pixels that lead to a false change and unreliability in Figure 6b. CD_SU is much better than CD_PC. There is an obvious improvement when the SU is implemented in the CD process in Figure 6c. Even so, the misclassifications of another class type, more or less, still affected the results. After the image is stacked, the performance of CD is improved in Figure 6d. The CD_SSU method can influence the occasional noise, and identification is presented to reduce erroneous judgments on failures, especially in W (blue). The acquired CD map in Figure 6e is closer to the reference. Due to the dividing process, CD_SDSU can measure a subtle but detectable change. It is found that most of the misclassifications of W, V and U in the former methods are fixed. In the end, Figure 6f is the most acceptable result, because it can fix all of the misclassifications. The proposed CD_SDSUVE reached the optimal performance considering the variations of the endmembers. The special treatment could be further in accord with the practical situation when the CD is implemented under the sub-pixel level. The statistics are consistent with the visual comparisons in Table 3. The proposed CD_SDSUVE provides the best results, with the maximum of OA and Kappa of 99.61% and 0.99, respectively.

**Table 3.** The accuracy statistics for the simulated data.

| Method | CD_PC | CD_SU | CD_SSU | CD_SDSU | CD_SDSUVE |
|--------|-------|-------|--------|---------|-----------|
| OA | 88.09% | 93.83% | 94.07% | 97.53% | 99.61% |
| Kappa | 0.76 | 0.91 | 0.91 | 0.96 | 0.99 |

*4.2. Real Multitemporal Image Data 1*

The real multitemporal image data 1 was chosen by the Sentinel-2 satellite, which had 13 bands with a spatial resolution of 20. The wavelengths ranged from 0.44 to 2.2 μm. Due to the low spatial resolution, three bands of 1, 9 and 10 of the images were removed. The subtracted images (400 × 400 pixels) of 2016 and 2018 in Hong Kong, China are shown in Figure 7a,b, respectively. The area is mainly covered by urban, water and vegetation. A visual analysis on Google Maps with high resolution may contribute to the definition of the change areas and, thus, change classes. Through comparative analyses, there are four major class changes, which are defined as the vegetation changed to urban (VU), the vegetation changed to bare soil (VB), the water changed to urban (WU) and the urban changed to water (UW). Figure 7c describes the typical changed land cover type spectral curves. There are four different colors in the figure: Green is VU, red is WU, blue is UW and magenta is VB.

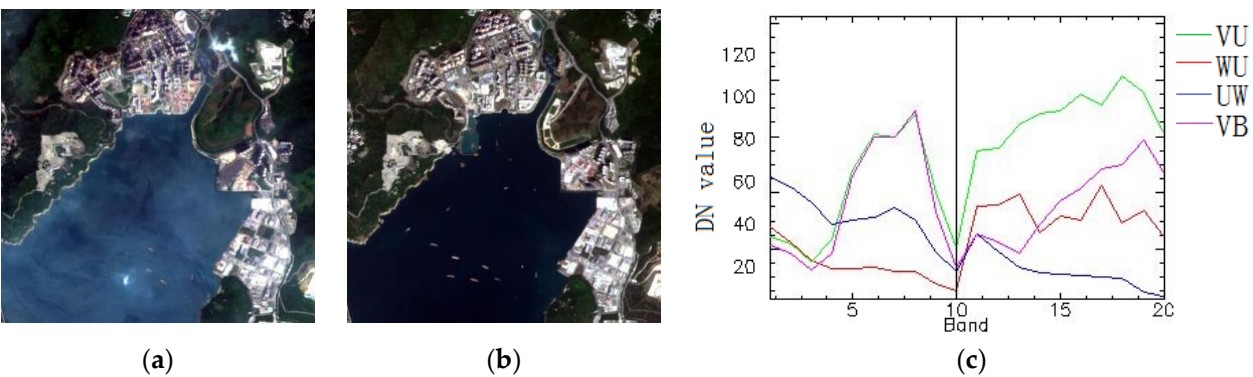

(**a**)                            (**b**)                            (**c**)

**Figure 7.** Real Multitemporal image data 1 (wavelength: R: 665 nm, G: 560 nm and B: 496 nm): (**a**) the original Sentinel-2 image in 2016, (**b**) the original Sentinel-2 image in 2018 and (**c**) four changed endmember spectral signature descriptions.

Like the simulated data, the stacked image was divided into four patches. Firstly, the initial endmember spectra were extracted in each patch image and collected to form four endmember groups: $G_1 \sim G_4$. All of the possible numbers according to a determined

land cover class in the different groups were acquired. All of the possible endmember spectral numbers in each endmember group are listed in Table 4. The number of the endmembers in the four groups were 17, 16, 14 and 18, respectively. After that, the changed and unchanged endmember spectra were discriminated using a threshold of 0.79. The similar endmember spectra were combined by SAM with 0.12. The EAR optimization was performed on the endmember pool, and the suitable endmembers were kept to establish a variable endmember matrix for the MEMSA model. Finally, the whole image was decomposed, and the fractional abundance images are shown in Figure 8a–h, respectively. It seems that there are significant differences among the different unchanged land cover types U, V and B. Additionally, the changed land cover types VU, VB, WU and UW all provided acceptable separations.

**Table 4.** The possible endmember spectral numbers in each group.

| Class Group | Number | | | | | | | |
| :---: | :---: | :---: | :---: | :---: | :---: | :---: | :---: | :---: |
| | U | V | B | W | VU | VB | WU | UW |
| $G_1$ | 5 | 3 | 1 | 1 | 3 | 2 | 2 | 0 |
| $G_2$ | 4 | 3 | 2 | 1 | 2 | 3 | 1 | 0 |
| $G_3$ | 3 | 2 | 1 | 2 | 1 | 2 | 2 | 1 |
| $G_4$ | 5 | 2 | 2 | 2 | 2 | 2 | 1 | 2 |

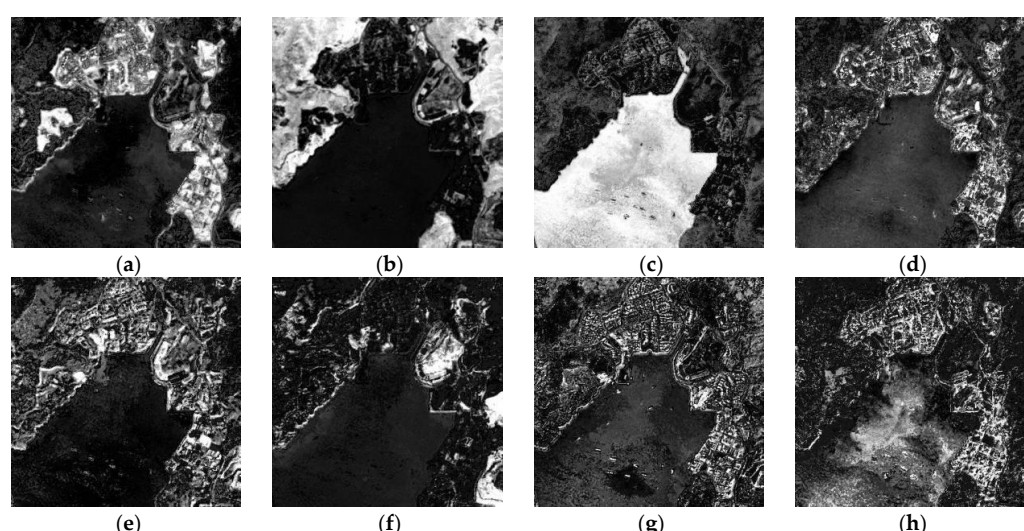

**Figure 8.** The abundances in the entire images: (**a**) urban (U), (**b**) vegetation (V), (**c**) water (W), (**d**) bare soil (B), (**e**) vegetation–urban (VU), (**f**) vegetation–bare soil (VB), (**g**) water–urban (WU) and (**h**) urban–water (UW).

All of the abundance maps were combined to produce the different change categories. Four colors were used for convenience describing the different changed land cover types. Green means VU, red is WU, blue is UW and magenta is VB. The unchanged land cover type was assigned as white. The reference land cover change map is shown in Figure 9a, which was produced by drawing the original images through visual interpretation in the same field on Google Maps with a high resolution. It was found that there were significant differences in the results by CD_PC comparing with the reference map in Figure 9b. Especially, the typical class type, bare soil, was not even detected. The changed and unchanged areas were mixed up to produce many misjudgments. For CD_SU, there was a distinct improvement. However, there were still a number of other possible uncertainties. For example, more urban shadows are misclassified for changing areas in Figure 9c. Some parts of the change WU and VB were not detected either. The CD_SSU in Figure 9d can provide more reliable and clearer edge information compared with CD_SU. The integrated

structure of the image is complete, and an enormous amount of details are effectively kept, especially in the boundary. The CD_SDSU is similar to CD_SSU, as it maintains a high correct rate in Figure 9e. However, there are still some misclassifications, especially in the area of the vegetation change to urban. Figure 9f shows the benefits of CD_SDSUVE by providing the most similar result to the reference image. The overall accuracy (OA) and Kappa value were used in the quantitative analysis of the mentioned CD methods. From Table 5, CD_PC is still the worst one, with the minimum of OA and Kappa of 78.38% and 0.44, respectively. CD_SU has a slightly improvement compared to CD_PC, a gain of 9.03% with OA over CD_PC. CD_SSU and CD_SDSU are similar. Their OA are 92.82% and 91.35%, respectively, and significantly better than CD_SU. The proposed CD_SDSUVE provides the best results, with the maximum of OA and Kappa of 93.26% and 0.66, respectively.

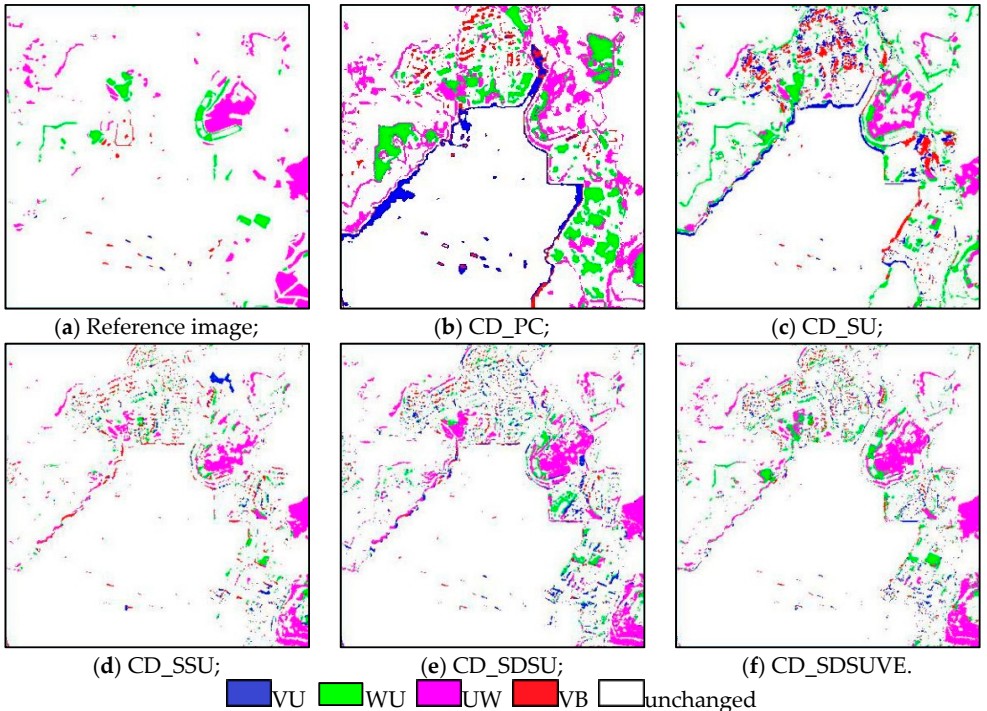

**Figure 9.** The change detection results of real data 1: (**a**) reference image, (**b**) CD_PC, (**c**) CD_SU, (**d**) CD_SSU, (**e**) CD_SDSU and (**f**) CD_SDSUVE.

**Table 5.** The accuracy statistics for real data 1.

| Method | CD_PC | CD_SU | CD_SSU | CD_SDSU | CD_SDSUVE |
|--------|-------|-------|--------|---------|-----------|
| OA | 78.38% | 87.41% | 92.82% | 91.35% | 93.26% |
| Kappa | 0.44 | 0.58 | 0.59 | 0.64 | 0.66 |

### 4.3. Real Multitemporal Image Data 2

The real multitemporal image data 2 was chosen by the Landsat TM with farmlands of Alabaster, AL, USA, which has undergone many changes. This distribution was more complicated than the former real data. A pseudo-color synthetic image with bands 4, 3, and 2 showed distant scenery around a suburb. The subtracted images with 900 × 900 pixels in 1988 and 2004 after co-registration are shown in Figure 10a,b, respectively. Compared with the two images at different times, there are three major classes: bare soil (B), rivulet (R) and agricultural crop (A). Through the original information, Google Maps can be linked with the test data for the definition of the change classes. The main change was that the agricultural crop changed to a rivulet (AR), the agricultural crop changed to bare soil (AB) and the bare soil changed to the agricultural crop (BA). First of all, the two temporal images

were stacked into an image pile, and then, the changed land cover type's spectral curves are as shown in Figure 10c. Green is AB, red is AR and blue is BA.

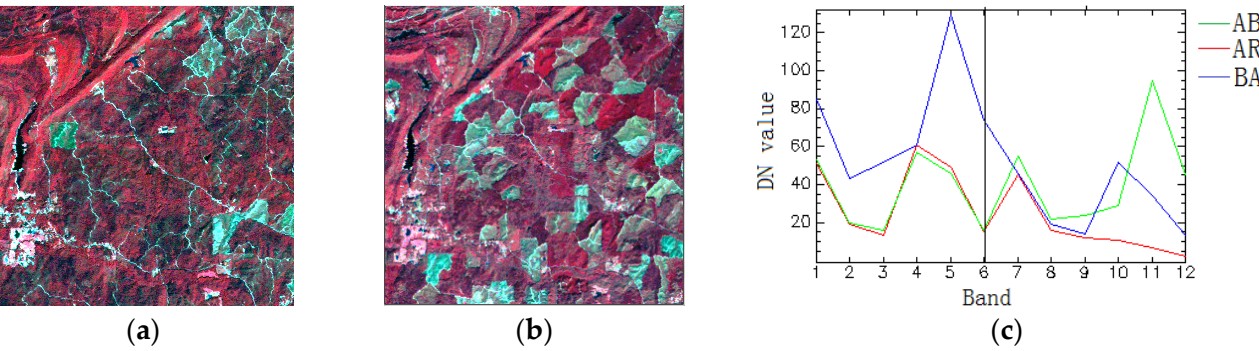

(a)                              (b)                              (c)

**Figure 10.** Real Multitemporal image data 2 (wavelength: R: 860 nm, G: 655 nm and B: 560 nm). (**a**) the original TM image in 2001, (**b**) the original TM image in 2009 and (**c**) three changed endmember spectral signature descriptions.

In this experiment, the divided scale was set as 3 due to the complexity and size of the image. There were nine endmember groups included by the extracted initial endmember spectra. Similarly, the possible numbers of the endmember spectra according to each class type in each group are described in Table 6. The number of the endmembers in the nine groups was 12, 12, 12, 14, 12, 12, 12, 12 and 12, respectively. After that, the changed and unchanged endmember spectra were discriminated using a threshold of 0.83. The similar endmember spectra were combined by SAM at 0.12. After comparison and discrimination, these spectra were classified as several land cover types. The EAR optimization was performed, and the suitable endmember spectra according to each land cover type were chosen. Next, the entire image was decomposed by MESMA according to the variable endmember combinations. The fractional abundance images are shown in Figure 11a–f, respectively. The main distribution land cover type was an agricultural crop. Meanwhile, bare soil and a rivulet could also be clearly separated. Especially, lots of bright values were shown in the agricultural crop–bare soil, which means that the abundance of this change land cover was the largest. The reference land cover change map was still concluded by manually drawing the original images through interpretation with a high resolution. Different colors were utilized to describe the changed and unchanged land cover types. White was unchanged, green was AR, red was AB and blue was BA. In contrast, the five methods had different results.

**Table 6.** The possible numbers of the endmember spectra in each group.

| Group \ Class | Number | | | | | |
|---|---|---|---|---|---|---|
| | **B** | **R** | **A** | **AR** | **AB** | **BA** |
| $G_1$ | 3 | 2 | 2 | 3 | 1 | 1 |
| $G_2$ | 2 | 3 | 2 | 2 | 1 | 2 |
| $G_3$ | 4 | 3 | 3 | 0 | 0 | 2 |
| $G_4$ | 5 | 3 | 3 | 0 | 2 | 1 |
| $G_5$ | 2 | 3 | 2 | 3 | 2 | 0 |
| $G_6$ | 3 | 3 | 2 | 1 | 1 | 2 |
| $G_7$ | 3 | 2 | 3 | 0 | 2 | 2 |
| $G_8$ | 2 | 3 | 3 | 0 | 2 | 2 |
| $G_9$ | 3 | 3 | 3 | 1 | 1 | 1 |

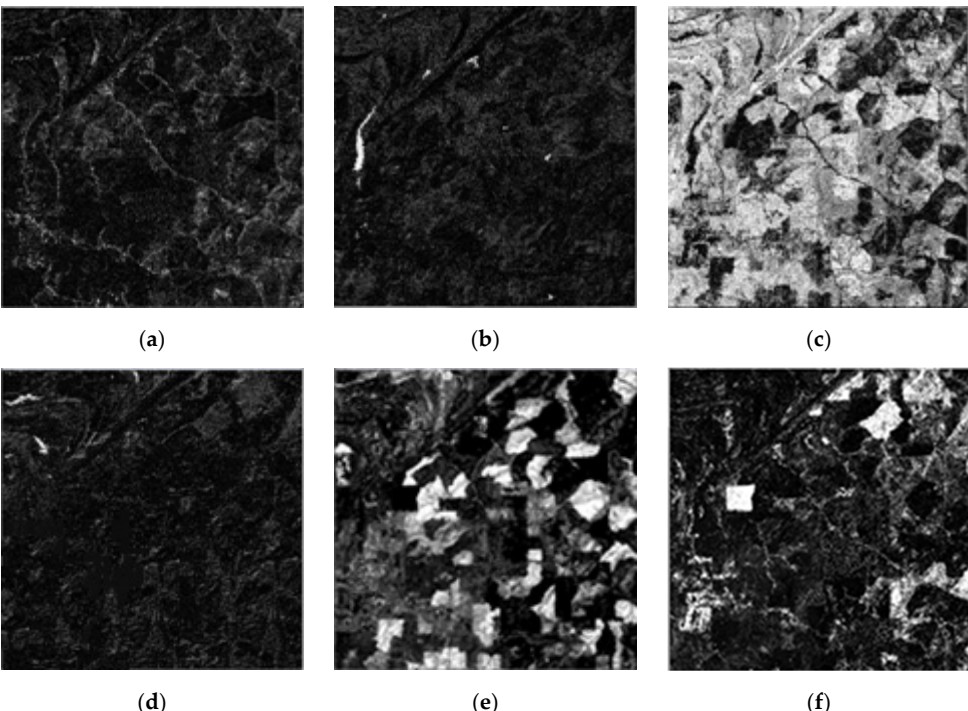

**Figure 11.** The abundances in the entire image: (**a**) bare soil (B), (**b**) rivulet (R), (**c**) agricultural crop (A), (**d**) agricultural crop–rivulet (AR), (**e**) agricultural crop–bare soil (AB) and (**f**) bare soil–agricultural crop (BA).

A reference map is shown in Figure 12a. The change detection results by different methods are shown in Figure 12b–h. For CD_PC, lots of changed areas, especially bare soil and agricultural crops, were not detected yet in Figure 12b. There were many misclassifications in the mixed area. Moreover, the shapes were commonly distorted, and the boundary of the changed land cover types were often blurred or smeared out by antialiasing. The CD_SU in Figure 12c is much better than CD_PC, but there are still a number of other possible errors and uncertainties. The CD_SSU in Figure 12d has a lot less misclassification areas, and it is more complete in the detection of agriculture-to-bare soil change areas compared with CD_SU. The CD_SDSU has a lot less misclassification areas and are more complete in the detection of agriculture-to-bare soil change areas in Figure 12e. However, many changes are still confused due to the existence of a same object with different spectra. Although the CD_SDSUVE also had misclassifications and undetected change areas, it outperformed the aforementioned methods in Figure 12f. The visual comparison was also proven by the accuracy statistics in Table 7. Performance was still evaluated with the OA and Kappa. In addition, the omission and commission errors are shown in Table 7. The number of pixels in the changed and unchanged area were 335,267 and 474,733, respectively. The OA and Kappa value of CD_PC were only 67.43% and 0.31, which were the worst results among all of the statistical results. CD_SU was better than the former, with a gain of 4.91% and 0.15. Additionally, CD_SSU and CD_SDSU were both increased to some degree. Their OA and Kappa values were 73.36% and 0.47 and 76.13% and 0.48, respectively. The proposed CD_SDSUVE had the best results, with an OA of 80.8% and a Kappa value of 0.56. Moreover, the omission and commission errors of CD_SDSUVE were lower than that of CD_PC, CD_SU, CD_SSU and CD_SDSU in all cases. The largest omission and commission errors in AR changed with CD_SU were 91.4% and 40.8%, respectively. All of the CD methods, except for CD_SDSUVE, always produced lots of missed and false changes, which led to large omission and commission errors.

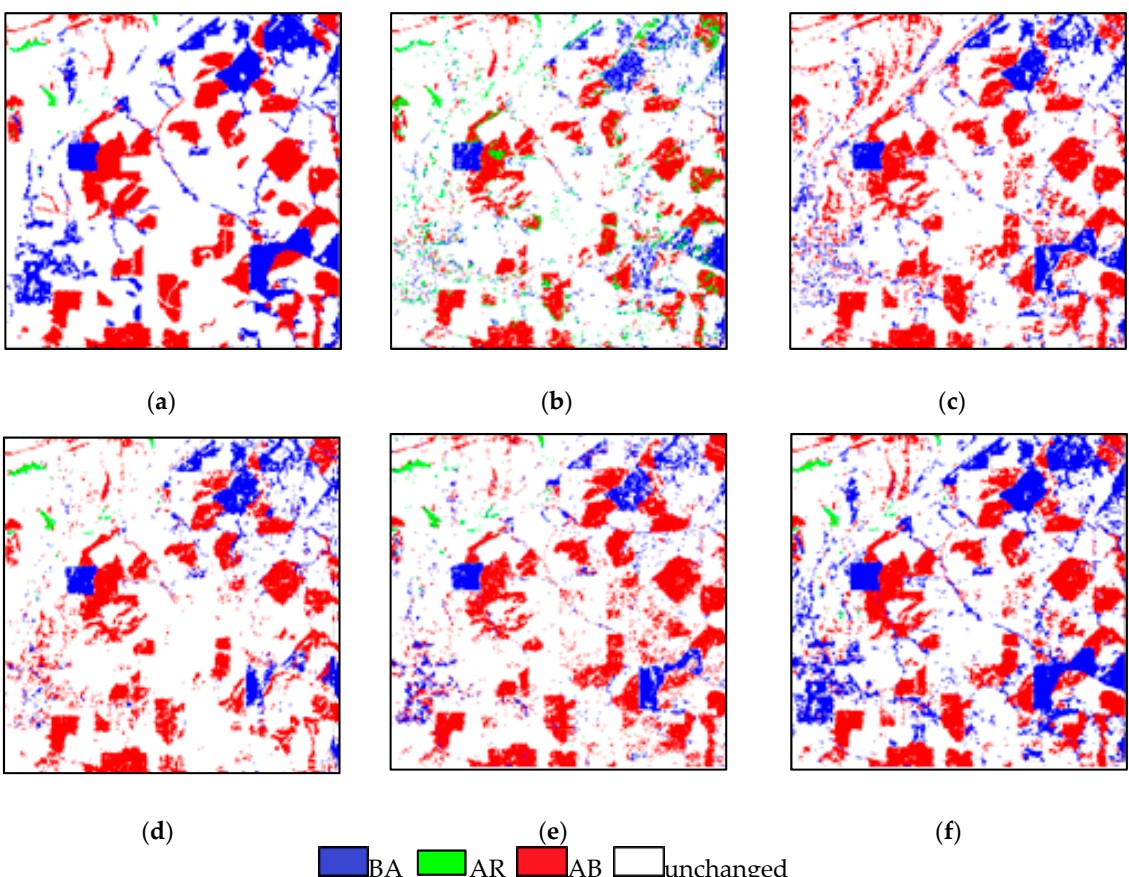

BA ■ AR ■ AB ■ unchanged □

**Figure 12.** The change detection results of real data 2: (**a**) reference image, (**b**) CD_PC, (**c**) CD_SU, (**d**) CD_SSU, (**e**) CD_SDSU and (**f**) CD_SDSUVE.

**Table 7.** The accuracy statistics for real data 2.

| Method | | CD_PC | CD_SU | CD_SSU | CD_SDSU | CD_SDSUVE |
|---|---|---|---|---|---|---|
| Omission Error | AR | 91.4% | 25.1% | 7.4% | 6.1% | 5.4% |
| | AB | 9.6% | 27.8% | 7.9% | 8.8% | 7.3% |
| | BA | 9.4% | 37.9% | 19.9% | 17.9% | 9.1% |
| Commission Error | AR | 40.8% | 18.5% | 16.1% | 16.4% | 13.5% |
| | AB | 22.2% | 22.9% | 17.5% | 18.7% | 15.3% |
| | BA | 16.3% | 25.5% | 13.4% | 22.8% | 10.0% |
| OA | | 67.43% | 72.43% | 73.36% | 76.13% | 80.85% |
| Kappa | | 0.31 | 0.46 | 0.47 | 0.48 | 0.56 |

## 5. Computational Complexity Analysis

The computational complexity of the five contrastive methods was different. CD_PC and CD_SU had the same process. Firstly, the multitemporal images should be unmixed/classified, which had to be processed individually. After that, the classification/unmixing images were compared to produce the CD result. CD_SSU did not need more space to store the intermediate images compared with the CD_PC and CD_SU. Since it stacked the multitemporal images and directly obtained the change information, less calculation times and space were needed. As for CD_SDSU, it had a dividing process to receive several patch images. The time and space cost of the algorithm were related to the patch number, image size and class number. The proposed CD_SDSUVE had the same part of the calculation process as the CD_SDSU. Additionally, the multiple endmember

selection contributed the most to the calculation times and space costs. The actual time costs of all the methods in the three experiments are listed in Table 8. The hardware and software conditions were as follows: a Lenovo laptop with Intel(R) Core (TM) i5-4200 CPU @2.4MHz, 8 Gb RAM, Windows 7 OS and Visual Studio 2010 IDE. CD_PC and CD_SU had similar times during the two experiments due to their similar computational processes. CD_SSU was the fastest algorithm, because the calculation time and space costs were both the smallest among all of the algorithms. CD_SDSU cost more time than CD_SSU, CD_PC and CD_SU, since it contained the image-dividing process. When the dividing number was large, each patch image produced more computational memory stress than the other nondividing algorithms. The proposed CD_SDSUVE was the slowest algorithm. In addition to the dividing process, the multiple endmember comparison and extraction led to more calculation time than CD_SDSU, but the time cost was still acceptable.

**Table 8.** Time costs of all the methods in the three experiments.

| Time (ms) | CD_PC | CD_SU | CD_SSU | CD_SDSU | CD_SDSUVE |
|---|---|---|---|---|---|
| Simulated data | 1410 | 1504 | 780 | 4126 | 7862 |
| Real data 1 | 1486 | 1513 | 981 | 4772 | 7985 |
| Real data 2 | 1517 | 1589 | 1240 | 5639 | 9623 |

Since the computational complexity of CD_SDSUVE was affected by the divided scale (s), it was necessary to analyze the effects of setting the parameter when running the CD_SDSUVE algorithm. For the three experimental data, the parameter of the divided scale s was set as 1, 2, 3 and 4, respectively. The number of patch images was 1, 4, 9 and 16, accordingly. The OA and time costs of the three experimental images when running the proposed algorithm are shown in Figure 13. As can be seen, the best performance appeared when the scale s was 2 and the time cost was 7862 in the simulated data. The differences in the accuracies among s = 2, 3 and 4 were very small, and the line chart of OA presented little vibration. A similar phenomenon also occurred in the real data 1. The best performance appeared when the scale s was 2 and the time cost was 7985. In addition, it was found that the time costs kept rising when the scale increased from 1 to 4. In the real data 2, the best performance appeared when the scale s was 3 and the time cost was 9623. Although the increase of the scale produced more patch images, the overall accuracy was not the more, the better. Thus, a conclusion can be drawn that CD_SDSUVE can reach an expectable result without more divided scales and time costs. It was critical for us to choose a reasonable divided scale to obtain the best results.

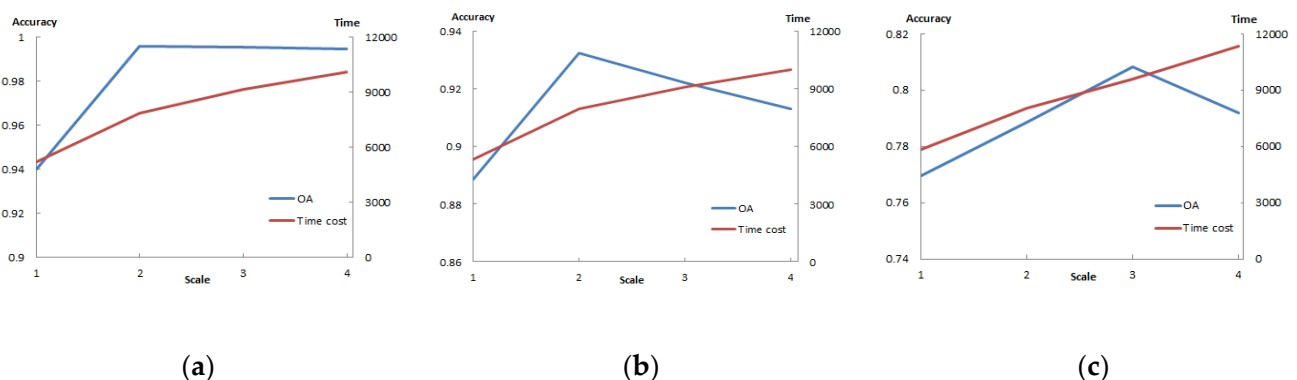

(a)             (b)             (c)

**Figure 13.** The OA and time costs of the three experiments with the different divided scales: (**a**) the line chart of the OA and time costs influenced in the simulated data, (**b**) the line chart of the OA and time costs influenced in the real data 1 and (**c**) the line chart of the OA and time costs influenced in the real data 2.

### 6. Discussion and Conclusions

This paper proposed a novel CD_SDSUVE method for change detection of multitemporal remote sensing images. There were two main key points to affect the CD results. The first one was to construct the endmember pool. After the stacking and dividing processes, the similar endmember spectra in the small, patched images were classified and combined. The specified endmember spectra according to land cover types were collected into the endmember pool. The second was to select the most suitable endmember spectrum from similar endmember spectra. An EAR indicator was proposed to obtain the minimum RMSE of the unmixing model. The optimal endmember spectrum was chosen by the calculated EAR indicator to decompose the mixed pixels. Four relevant state-of-the-art algorithms of CD_PC, CD_SU, CD_SSU and CD_SDSU were compared with CD_SDSUVE. Although the proposed method tended to consume more time than the other methods, the time costs were still acceptable. The simulated and real experimental results confirmed the superiority of the proposed method, whereby the accuracy and visual effects of the resulting change detection maps were significantly improved.

From the theoretical analysis and the practical experimental results, we can conclude the following: (1) The change detection problem can be effectively resolved in an unsupervised framework. The entire CD process is under a unified framework, and no training samples are needed. Furthermore, image dividing based on stacked images can detect much smaller changes, which increases the number and types of endmembers and is more conducive to spectral unmixing. (2) The multiple endmember spectral mixture analysis was more suitable for decomposing the mixed pixels in this situation. The number and type of endmember spectra could be adjusted according to the real situation of the patch images. This means the use of multi-endmember information where available is considered by using the variability of the endmembers. Future works should focus on the improvement of endmember extractions in patch images and suitable endmember combinations for spectral unmixing.

**Author Contributions:** K.W. and D.S. proposed the algorithm, conceived and designed the experiments and performed the experiments; K.W., T.C., Y.X., D.S. and H.L. provided article revision opinions and K.W. wrote the paper. All authors have read and agreed to the published version of the manuscript.

**Funding:** This research was funded in part by the Natural Science Foundation of China (62071439); the National Defense Pre-Research Foundation of China during the 13th Five-Year Plan Period: the High Spectral Resolution Infrared Space-Based Camera and the Applied Technology under grant D040104 and the Military and Civilian Integration for Marine Comprehensive Survey and Application of the Maritime Silk Road under Grant (2019061160).

**Institutional Review Board Statement:** Not applicable.

**Informed Consent Statement:** Not applicable.

**Data Availability Statement:** The data presented in this study are available on request from the corresponding author.

**Acknowledgments:** The authors would like to thank the anonymous reviewers and associate editor for their valuable comments and suggestions to improve the quality of the paper.

**Conflicts of Interest:** The authors declare no conflict of interest. The founding sponsors had no role in the design of the study; in the collection, analyses or interpretation of the data; in the writing of the manuscript or in the decision to publish the results.

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
