# Peer review of "A Novel Change Detection Approach Based on Spectral Unmixing from Stacked Multitemporal Remote Sensing Images with a Variability of Endmembers"

_remotesensing, doi:10.3390/rs13132550_

Round 1

Reviewer 1 Report

This manuscript proposes a method for change detection. However, it requires a lot of modifications. The reviewers’ comments are as follow.

  • Abstract contains a lot of details. Please shorten it and describe main information.
  • CD-SU has been mentioned as general method in Introduction. In experimental section please insert the reference number for compared methods.
  • CD-SSU and CD-SDSU have not been explained in Introduction Section.
  • Please highlight the novelty as bullet in Introduction Section. In the current version, it is not clear.

Reviewer 2 Report

Change detection approaches are very important especially to determine the LULC changes with high overall accuracy. The subject of the manuscript is very important but there are some points that the authors should provide detailed explanation about them. Because of this reason my suggestion about the manuscript is reconsider after major revision. 

1) The authors should improve the abstract section by adding statistical results. 

2) The author should re-organized the introduction section by adding more references. 

3) The authors should provide detailed information about parameter s (divided scale parameter). 

4) What is N-FINDR algorithm. Please add information about it. 

5) The author should provide detailed information about threshold value for d ( the absolute difference value). I think that visual interpretation is not the best way to determine the threshold value for a novel approach. 

6) The authors should re-think about the section 3.5. I think there is no need for this section.

7) Till section 4 there is no information about the selected data set. The authors should provide information about used data and their properties before analysis. 

8) What is the meaning of simulated Landsat TM image ? The authors should provide information about this. 

9) The authors should add legend to all classification results (maps) and spectra graphics such as figure 4 c, figure 6, figure 7 c, figure 9, figure 10.c, figure 12. 

10) The authors should provide more detailed information about reference maps

11) The authors should provide error matrix for each accuracy assessment ( not  only overall accuracy and Kappa statistics ). 

12) The authors should improve the discussion section. 

Reviewer 3 Report

This paper addresses the change detection problem by spectral unmixing considering variability of endmembers on the stacked and divided hyperspectral images. A series of experiments have conducted to demonstrate the merits of the proposed method. Overall, the current version of this work is well written and well organized, but of limited feasibility and contribution. As such, I cannot suggest it for publication in remote sensing.

Basically, I have the following main comments.

  1. Feasibility. The proposed algorithm is overly dependent on hyperparameters, such as endmember number, the appropriate threshold value for the spectral absolute differences, the spectral angle, EAR, etc. The author did not give the setting method of these hyperparameters. If these hyperparameters are all empirically set, then the feasibility of the proposed method is very low.
  2. Experimental comparison. The comparison methods chosen in the paper does not indicate the reference. Although such compared method may be considered as the state-of-the-art method,  I think this is not enough to support the merits of the proposed method.
  3. Reproducibility of the experiment. The hyperparameter setting values are not set in the paper for either the proposed method or the comparison method, which makes the experiments less reproducible.
  4. There are also some unclear points in the text. For example, why stacking first and then splitting, how to obtain changes by combining, such as VW, UV, WU, etc. In line 328 above Figure 7, there should be four classes. Then it should be at least 6 classes without considering the reverse change, why 4 classes? What are the rules for the definition of change? How to define it?

Reviewer 4 Report

See attached file.

Reviewer 5 Report

The paper presents a novel change detection approach based on SU with variability of endmembers.The contribution of the work is quantified, followed by different experimtental results comparing the fundamental CD methods. I generally enjoyed their approach and it gives value to the literature as long as the following are satisfied. The authors describe the method, and even use real data. The reference map is quite essential for the comparison with the different methods. The authors need to emphysize the how to get the reference map. Generally, I believe this paper is well written. There are still some minor problem, e.g. the wording is not very standard English; the typoes or symbol inconsistency (e.g. Line 112-113, Line 184-187); but overall it is still mostly readable.

Author Response

The paper presents a novel change detection approach based on SU with variability of endmembers. The contribution of the work is quantified, followed by different experimtental results comparing the fundamental CD methods. I generally enjoyed their approach and it gives value to the literature as long as the following are satisfied. The authors describe the method, and even use real data. The reference map is quite essential for the comparison with the different methods. The authors need to emphysize the how to get the reference map. Generally, I believe this paper is well written. There are still some minor problem, e.g. the wording is not very standard English; the typoes or symbol inconsistency (e.g. Line 112-113, Line 184-187); but overall it is still mostly readable.

R: Thank you for your comment. In the simulated data, the reference land cover change map is produced by these man-made exchanged areas. In the real data, the Google Maps can be linked. The reference land cover change map is produced by drawing the original images through the visual interpretation in the same field aid. Visual analysis on Google Maps with high resolution may contribute to the definition of change areas and, thus, change classes. Some contents have been added to the manuscript. We have fixed all the mentioned problems (e.g. Line 112-113, Line 184-187) and proofread the manuscript.

Reviewer 6 Report

This paper proposed a novel change detection approach based on spectral unmixing by using time series data. Overall, the structure of this paper is well organized, and the presentation is clear. However, there are still some crucial problems that need to be carefully addressed before a possible publication. More specifically,

  1. A deep literature review should be given, particularly regarding hyperspectral unmixing. Therefore, the reviewer strongly suggests discussing and analyzing some advanced and latest works by citing the following papers, e.g. ““An Augmented Linear Mixing Model to Address Spectral Variability for Hyperspectral Unmixing.”
  2. How about the computational complexity?
  3. Please clarify the differences between the proposed method and existing learning-based networks.
  4. The contributions of this paper are not so clear to the reviewer. Please clarify which one is existing and which one is your own?

Round 2

Reviewer 1 Report

Thank you for your revision work. However, the manuscript requires some modifications.

  • In order to show the importance of your method, please insert another table and experiment and compare your method with change detection papers having recently been published with mentioning the reference number of papers.

Author Response

Thank you for your revision work. However, the manuscript requires some modifications. In order to show the importance of your method, please insert another table and experiment and compare your method with change detection papers having recently been published with mentioning the reference number of papers.

Response (R):  Thank you for your comment. We do understand your concerning about this point. As we known, some typical learning-based CD methods mentioned in the reference papers, just like k-means clustering, change vector analysis (CVA), etc., can be seen as pixel level models. These kinds of the techniques always assume that a pixel is according to a single class. In contrast, the proposed method allows one to understand, in detail, the spectral composition of a pixel by spectral unmixing, thus implementing CD at the subpixel level. We just compared the proposed CD_SDSUV with traditional CD_PC, CD_SU, CD_SSU and CD_SDSU in the experiment, because the emphasis of the manuscript is whether the consideration of stacking, dividing and VE (variability of endmembers) can improve the accuracy of the CD result. That is, the comparison is focused on the difference between CD models with or without these additional processes. This is the main contribution of the manuscript; as far as we know, there is no related research in this area. Through the experiment, CD_PC always provides the worst result because it is the only pixel-level CD method. There is an obvious improvement when the SU is implemented in the CD process. After the image is stacked, divided and VE consideration, the CD_SDSUVE can reach optimal performance. The special treatment could be further accord with the practical situation when the CD is implemented under a sub-pixel level. Although methods in the mentioned reference papers were not presented in our experiment, the key points of the proposed algorithm can be still addressed. In the future, the advanced unsupervised multi-temporal unmixing approaches will be developed and compared.

Reviewer 3 Report

This manuscript was modified appropriately and some instructive improving had been done. Authors are fulfilled all my concern comments in revision version of the paper. We are willing to recommend its acceptance.

Author Response

Thank you for your comment.